# AGO1 regulates pericentromeric regions in mouse embryonic stem cells

Madlen Müller[1,2] , Tara Fäh[1], Moritz Schaefer[1,2] , Victoria Hermes[1], Janina Luitz[1], Patrick Stalder[1,2], Rajika Arora[1] , Richard Patryk Ngondo[1] , Constance Ciaudo[1]

**Argonaute proteins (AGOs), which play an essential role in cytosolic post-transcriptional gene silencing, have been also reported to function in nuclear processes like transcriptional activation or repression, alternative splicing and, chromatin organization. As most of these studies have been conducted in human cancer cell lines, the relevance of AGOs nuclear functions in the context of mouse early embryonic development remains uninvestigated. Here, we examined a possible role of the AGO1 protein on the distribution of constitutive heterochromatin in mouse embryonic stem cells (mESCs). We observed a specific redistribution of the repressive histone mark H3K9me3 and the heterochromatin protein HP1α, away from pericentromeric regions upon *Ago1* depletion. Furthermore, we demonstrated that major satellite transcripts are strongly up-regulated in *Ago1*_KO mESCs and that their levels are partially restored upon AGO1 rescue. We also observed a similar redistribution of H3K9me3 and HP1α in *Drosha*_KO mESCs, suggesting a role for microRNAs (miRNAs) in the regulation of heterochromatin distribution in mESCs. Finally, we showed that specific miRNAs with complementarity to major satellites can partially regulate the expression of these transcripts.**

## Introduction

The miRNA pathway is crucial in regulating early embryonic development and differentiation in vivo and in vitro (DeVeale et al, 2021). MiRNAs can fine-tune gene expression throughout early embryonic development at the post-transcriptional level. MiRNA precursors are processed into ~22-nt long mature miRNAs by two consecutive cleavage steps conducted by the RNAse III enzyme DROSHA in the nucleus, and DICER, in the cytoplasm (Bodak et al, 2017). Mature miRNAs are loaded into Argonaute (AGO) proteins, which are key components of the RNA-induced silencing complex.

They guide the RNA-induced silencing complex to partially complementary target sequences leading to the translational inhibition of these targets (Bartel, 2018).

In mice, there are four AGO proteins (AGO1-4), but only AGO1 and AGO2 are detectably expressed during early embryonic development, with AGO2 being substantially more abundant (Lykke-Andersen et al, 2008; Boroviak et al, 2018; Müller et al, 2020). Whereas *Ago2*-deficient mice die at a post-implantation stage, because of severe developmental defects (Liu et al, 2004; Alisch et al, 2007; Morita et al, 2007; Cheloufi et al, 2010), *Ago1,3,4*-deficient mice are viable (Modzelewski et al, 2012; Van Stry et al, 2012).

Mouse embryonic stem cells (mESCs), which are derived from the inner cell mass of the blastocyst, are a powerful tool to study early embryonic development in vitro. These cells are pluripotent and can differentiate into the three embryonic germ layers. As observed in vivo, mESCs only express AGO1 and AGO2 proteins (Lykke-Andersen et al, 2008; Boroviak et al, 2018; Müller et al, 2020). MESCs deficient for either AGO1 or AGO2 are viable, can exit from pluripotency and differentiate into the three embryonic germ layers (Ngondo et al, 2018).

In addition to their major role in the cytoplasmic miRNA pathway, several studies have reported noncanonical functions of the AGO proteins in the nucleus (Meister, 2013; Gagnon et al, 2014a; Li et al, 2020). AGO2 was shown to shuttle into the nucleus with the help of TNRC6A (Nishi et al, 2013). In the nucleus, guided by small RNAs (smRNAs), both AGO1 and AGO2 have been shown to localize to promoter regions and reinforce the recruitment of chromatin modifiers leading to either transcriptional activation or silencing (Janowski et al, 2006; Kim et al, 2006; Li et al, 2006; Hu et al, 2012; Cho et al, 2014; Portnoy et al, 2016). AGO1 was also found to be enriched at promoters of actively transcribed genes, where it interacts with RNA Polymerase II (RNA PolII) (Huang et al, 2013). Furthermore, AGO1 was found to localize to enhancer regions, which was dependent on a species of RNA called enhancer RNAs (Alló et al, 2014; Shuaib et al, 2019). In addition, interaction of AGO1 with enhancers was shown to be crucial for maintenance of 3D chromatin organization and more recently to control myogenic differentiation (Shuaib et al, 2019;

[1]Swiss Federal Institute of Technology Zurich, Institute of Molecular Health Sciences (IMHS), Chair of RNAi and Genome Integrity, Zurich, Switzerland   [2]Life Science Zurich Graduate School, University of Zürich, Zürich, Switzerland

Correspondence: cciaudo@ethz.ch
Richard Patryk Ngondo's present address is Institut de Biologie Moléculaire des Plantes UPR-CNRS 2357, Strasbourg, France.

Fallatah et al, 2021). Finally, AGO1 has been implicated in alternative splicing events, taking place within the nucleus (Ameyar-Zazoua et al, 2012; Alló et al, 2014; Agirre et al, 2015). Of note, most of these chromatin-associated functions have been described mainly for AGO1, whereas AGO2 was reported to be involved in double-strand break repair (Gao et al, 2014; Wang & Goldstein, 2016).

Most of the aforementioned AGOs functions were described in human cancer cell lines and have not been studied during early embryonic development. Only few studies reported phenotypes associating the AGO proteins with other functions in mESCs (Sarshad et al, 2018; Kelly et al, 2019; Shivram et al, 2019). For instance, Kelly et al (2019) identified that TGF-β pathway targets are up-regulated upon AGO2 depletion in mESCs, due to a lack of miRNA repression. The up-regulation of these targets additionally correlated with decreased levels of the repressive histone mark, H3K27me3 (Kelly et al, 2019). Interestingly, we also observed a specific loss of H3K27me3 mark in *Ago1&2*_KO mESCs (Mueller et al, 2021 *Preprint*). Repressive histone marks are important for the formation of heterochromatin, which is localized to distinct territories within the nucleus (Akhtar & Gasser, 2007; Solovei et al, 2009).

In this study, we aimed to explore the link between AGO1 and heterochromatin by assessing both the amount and the distribution of constitutive heterochromatin in *Ago1* mutant mESCs. We observed a specific redistribution of the repressive histone mark H3K9me3 and, the heterochromatin protein HP1α away from pericentromeric regions in *Ago1*_KO mESCs. Furthermore, these regions are characterized by AT-rich tandem repeats known as major satellite sequences. We demonstrated that major satellite transcripts are strongly up-regulated in *Ago1*_KO mESCs. Nevertheless, we did not observe any changes in integrity of the pericentromeric region at the DNA level. Importantly, these phenotypes were rescued upon the reintroduction of AGO1 in the mutant cells. These results prompted us to investigate the underlying molecular mechanism by which AGO1 might regulate major satellite transcripts. We first demonstrated the involvement of miRNAs by observing a similar redistribution of H3K9me3 and HP1α in *Drosha*_KO mESCs. Using computational analyses and molecular approaches, we also found that AGO1, loaded with miR-30a, d, e-3p, might contribute partially to the regulation of major satellite transcripts. Overall, our results demonstrate for the first time a novel role for AGO1 in regulating major satellite transcripts and localization of H3K9me3 and HP1α at pericentromeres in mESCs.

## Results

### *Ago1*-depletion affects the distribution of H3K9me3 and HP1α at pericentromeric regions

Only the H3K27me3 heterochromatin mark, but not H3K9me3, was previously observed to be strongly down-regulated in Argonaute mutant mESCs (Kelly et al, 2019; Mueller et al, 2021 *Preprint*). Heterochromatin is localized to specific nuclear territories in mammalian cells (Akhtar & Gasser, 2007; Solovei et al, 2009). H3K9me3, in particular, is enriched at pericentromeric constitutive heterochromatin regions in mammals, which can be found at the centromeres.

Constitutive heterochromatin at centromeres is required for proper sister chromatid cohesion and chromosome segregation (Bernard et al, 2001; Nonaka et al, 2002; Guenatri et al, 2004; Houlard et al, 2006; Probst & Almouzni, 2008; Probst et al, 2009). Pericentromeric domains from several chromosomes are known to cluster together within interphase to form chromocenters (Guenatri et al, 2004; Probst & Almouzni, 2008). Chromocenters are easily visible by fluorescence microscopy with a brighter DAPI stain (Guenatri et al, 2004; Probst & Almouzni, 2008). AGO1 has been previously linked to chromatin-associated functions in mammalian cells (Huang et al, 2013; Alló et al, 2014; Shuaib et al, 2019). To study whether AGO1 might be important for constitutive heterochromatin localization in mESCs, we used two *Ago1*_KO mESC lines generated using a paired CRISPR-Cas9 approach (Wettstein et al, 2016). The first *Ago1*_KO1 mESC line was obtained from a previous study (Ngondo et al, 2018) and the second *Ago1*_KO2 mESCs line was newly generated and validated for the absence of AGO1 expression (Fig S1A and B). As previously observed in *Ago1&2*_KO mESCs (Mueller et al, 2021 *Preprint*), the total amount of H3K9me3 histone mark as assessed by Western blotting (WB) was similar in WT versus *Ago1*_KO mESCs (Fig S1C). To go further, we analyzed the nuclear localization of H3K9me3 by indirect Immuno-fluorescence (IF) and observed colocalization of H3K9me3 with DAPI-rich regions in WT mESCs (Figs 1A and S1D). Surprisingly, this colocalization of H3K9me3 with DAPI-rich regions, was strongly reduced in *Ago1*_KO mESCs (Fig 1A).

To strengthen these observations, we performed H3K9me3 chromatin immunoprecipitation followed by quantitative PCR (ChIP-qPCR) in WT versus *Ago1*_KO mESCs and successfully assessed the enrichment at known heterochromatic loci (Karimi et al, 2011; Ngondo et al, 2018) over a control region (Fig S1E and Table S1). We also compared the H3K9me3 enrichment at major satellites sequences in both WT and *Ago1*_KO, and observed that upon *Ago1* depletion, around 50% of H3K9me3 is lost at these sites (Fig 1B and Table S1).

H3K9me3 is deposited at pericentromeric heterochromatin regions by the methyltransferase SUV39H1/2, which is recruited there by HP1α (also known as CBX5) (Bannister et al, 2001; Lachner et al, 2001; Hyun et al, 2017). Therefore, we assessed the colocalization of HP1α with DAPI-rich regions in WT and *Ago1*_KO mESCs by IF (Fig 1C). Similarly, we observed a significant redistribution of HP1α in *Ago1*_KO mESCs, away from the pericentromeric regions (Fig 1C). In addition, we noted a slight increase in HP1α protein expression in *Ago1*_KO mESCs compared with WT cells (Fig S1F).

In conclusion, we observed a redistribution of both the repressive histone mark H3K9me3 and the heterochromatin protein HP1α, away from pericentromeric regions in *Ago1*_KO mESCs.

### AGO1 complementation rescues the distribution of H3K9me3 and HP1α at pericentromeric regions

To determine, whether the redistribution of H3K9me3 away from pericentromeric regions is specific to the loss of AGO1, we aimed to complement our *Ago1*_KO mESCs. We transfected the *Ago1*_KO2 mESC line with a vector expressing N-terminally HA-tagged AGO1. This vector additionally contains two selection markers, a GFP and a puromycin resistance gene. Cells expressing AGO1 were selected for a week for puromycin resistance, followed by FACS sorting for GFP (Fig S2A). Finally, we verified HA-AGO1 expression in the GFP sorted

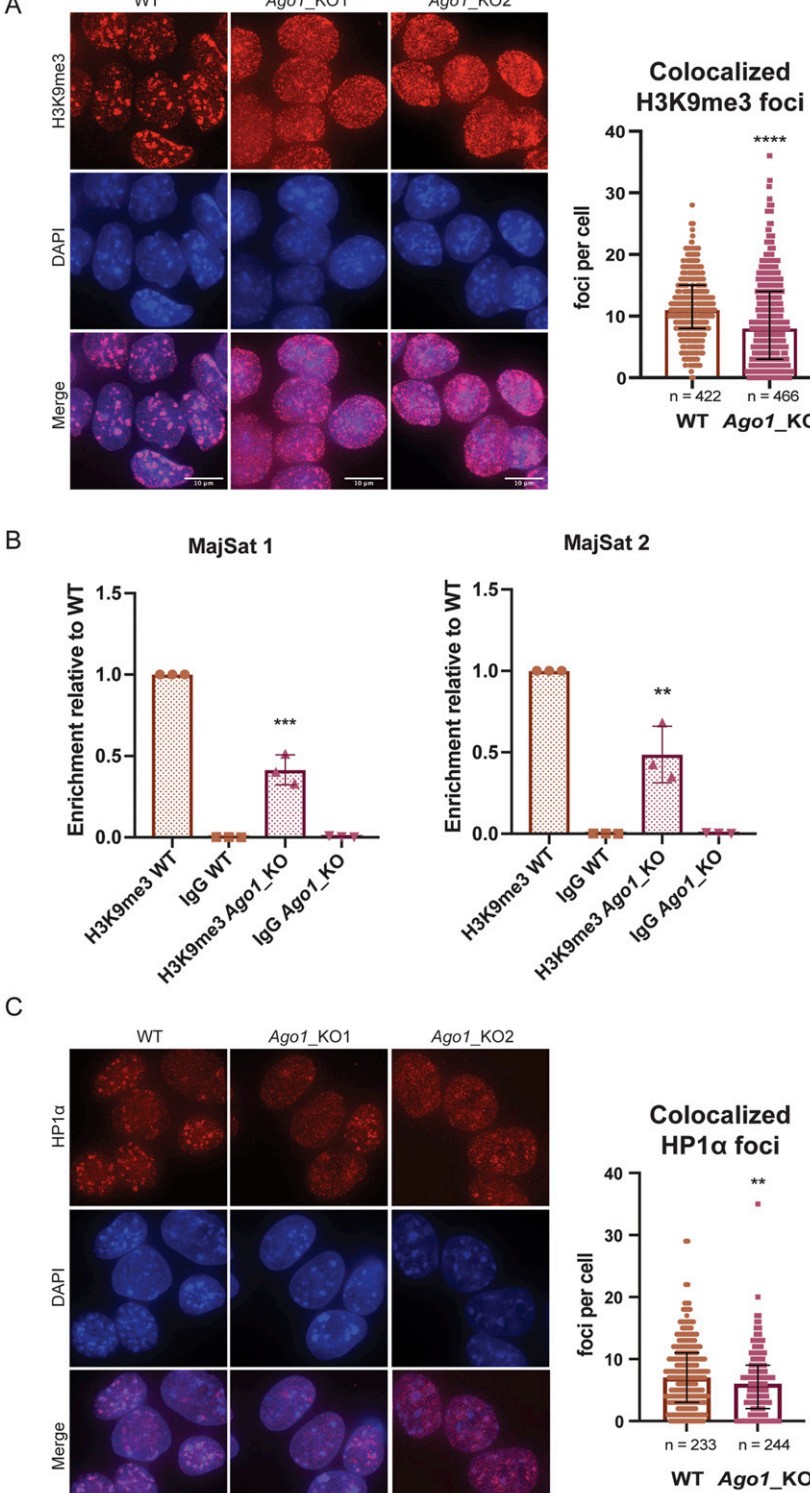

**Figure 1. Distribution of H3K9me3 and HP1α at pericentromeric regions in WT versus *Ago1*_KO mouse embryonic stem cells (mESCs).**
**(A)** Left: representative IF images of H3K9me3 in WT and *Ago1*_KO mESCs. Scale bar = 10 μm. Right: quantification of foci counts for H3K9me3 that colocalizes with DAPI regions in WT and *Ago1*_KO mESCs. Because of the bimodal distribution of H3K9me3 foci in *Ago1*_KO mESCs, the graph shows the median distribution with the interquartile range. **** = *P*-value < 0.0001, Mann–Whitney test for n = 3 independent experiments. **(B)** ChIP-qPCR in WT and *Ago1*_KO mESCs. Pull-downs were performed with an antibody against H3K9me3 and a control IgG antibody. qPCR has been performed on major satellite primer set 1 and 2 (Table S1). The enrichment was calculated over input and represented relative to the WT H3K9me3 pull-down. *** = *P*-value < 0.001 and ** = *P*-value < 0.01, unpaired *t* test for n = 3 independent experiments. The *Ago1*_KO1 and *Ago1*_KO2 have been combined in this experiment. IgG error bars clipped at axis limit. **(C)** Left: representative IF images of HP1α in WT and *Ago1*_KO mESCs. Scale bar = 10 μm. Right: quantification of foci count for HP1α that colocalizes with DAPI regions in WT and *Ago1*_KO mESCs. The graph shows the median distribution with the interquartile range. ** = *P*-value < 0.01, Mann–Whitney test for n = 3 independent experiments. For the quantification *Ago1*_KO1, and *Ago1*_KO2 were combined.

polyclonal cell population (mixed cell population) by IF and WB, and observed a partial rescue of AGO1 levels in the complemented cells (Figs 2A and S2B). We next tested whether the reintroduction of AGO1 could rescue the distribution of H3K9me3 foci at DAPI-rich regions and performed a co-staining for H3K9me3 and HA in WT, *Ago1*_KO and the *Ago1*_KO + HA-AGO1 cells (Fig 2A). The co-staining with HA allowed us to select cells re-expressing AGO1 at a proper level in the polyclonal population to perform the quantification.

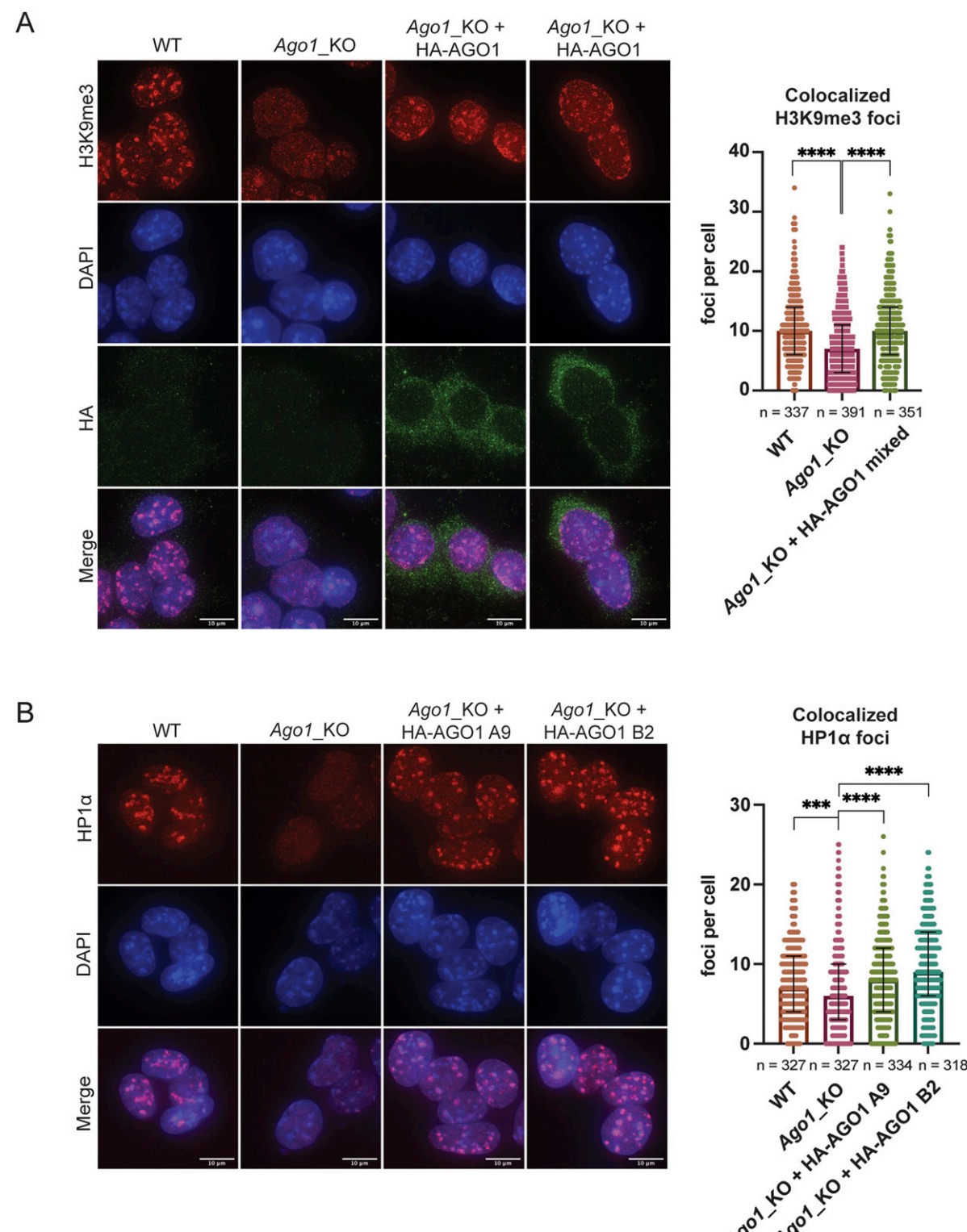

**Figure 2. AGO1 complementation rescues the distribution of H3K9me3 and HP1α at pericentromeric regions in *Ago1*_KO Mouse Embryonic Stem Cells (mESCs).**
**(A)** Left: representative IF images of H3K9me3 in WT, *Ago1*_KO and two representative images of *Ago1*_KO + 3x HA-AGO1 mixed population mESCs. Scale bar = 10 μm. Right: quantification of foci count for H3K9me3 that colocalizes with DAPI regions in WT, *Ago1*_KO, and *Ago1*_KO + 3x HA-AGO1 mixed population mESCs. The graph shows the median distribution with the interquartile range. **** = *P*-value < 0.0001, Mann–Whitney test for n = 3 independent experiments. **(B)** Left: representative IF images of HP1α in WT, *Ago1*_KO, and *Ago1*_KO + 3x HA-AGO1 single clones A9 and B2. Right: quantification of foci count for HP1α that colocalizes with DAPI regions in WT, *Ago1*_KO, and *Ago1*_KO + 3x HA-AGO1 single clones A9 and B2. The graph shows the median distribution with the interquartile range. **** = *P*-value < 0.0001, *** = *P*-value < 0.001, Mann–Whitney test for n = 3 independent experiments.

We observed a significant rescue of the H3K9me3 distribution at the pericentromeric regions upon reintroduction of AGO1 in *Ago1*_KO mESC line (Fig 2A).

Because of the nuclear pre-extraction step necessary for proper visualization of HP1α by IF, we were not able to simultaneously assess the distribution of HP1α along with HA-AGO1 in the complemented polyclonal population. To solve this issue, we selected monoclonal clones using GFP as a marker by FACS sorting (Fig S2C). Two clones (B2 and A9) were kept for further analysis as they expressed HA-AGO1 at similar (B2) or at a higher level (A9) than the complemented polyclonal population of cells previously analyzed (Fig S2D). We then assessed the distribution of HP1α in WT, *Ago1*_KO and *Ago1*_KO complemented clones by IF and, observed a similar significant rescue of the HP1α distribution at the pericentromeres upon reintroduction of AGO1 in *Ago1*_KO mESC single clones (Fig 2B).

In conclusion, the reintroduction of AGO1 in *Ago1*_KO mESCs partially rescue the mislocalization of both H3K9me3 and HP1α at pericentromeres.

## Major satellite transcripts are up-regulated in *Ago1*_KO mESCs

In mouse, pericentromeric heterochromatin regions are characterized by AT-rich tandem repeats, known as major satellite repeat sequences. Major satellites consist of 234 bp tandem repeat sequences that can stretch over several kilobases (Guenatri et al, 2004; Komissarov et al, 2011). The minor satellite sequences adjacent to the major satellites are localized to the centromeric part of the chromosome (Fig S3A) (Vissel & Choo, 1989; Guenatri et al, 2004). Even though pericentromeric regions are marked by repressive heterochromatin marks, pericentromeric transcripts, such as major satellite transcripts, have been previously reported to be expressed in vivo during mouse early development and also in vitro in mESCs (Fig S3A) (Rudert et al, 1995; Lehnertz et al, 2003; Probst et al, 2010).

To assess whether the depletion of *Ago1* affects major satellite transcripts, we performed an RNA FISH (Fig 3A). We observed a stronger signal and significantly more foci corresponding to major satellites in *Ago1*_KO compared with WT mESCs (Figs 3A and S3A). To confirm that the increase in foci number per cell corresponding to transcripts derived from major satellites is specific to the loss of AGO1 in mESCs, we performed RNA FISH in the previously derived single complemented clones A9 and B2. We observed that the reintroduction of AGO1 in *Ago1*_KO mESCs could also partially rescue the RNA FISH signal in the single complemented clones (Fig 3A). To better quantify the amount of major satellite transcripts in all cell lines, we then measured their relative expression using a stringent RT-qPCR protocol in WT, *Ago1*_KO, and Ago1_KO complemented clones (see the Materials and Methods section and Fig S3B). We detected a significant up-regulation of major satellite mRNAs in the *Ago1*_KO mESCs using two independent primer pairs and this up-regulation was decreased by half upon reintroduction of AGO1 in *Ago1*_KO mESC clones (Fig 3B).

Finally, to better understand whether the up-regulation of major satellite transcripts was linked to changes at the chromatin level, we analyzed the IF images for the H3K9me3 staining and quantified the number of DAPI foci (chromocenters) in WT versus *Ago1*_KO

mESCs. We observed no decrease in DAPI foci formation in *Ago1*_KO compared with WT mESCs (Fig S3C). In addition, we also performed DNA FISH for the major satellite repeats in these two cell lines. The major satellite DNA FISH signal was similar between WT and *Ago1*_KO mESCs and we did not detect a more dispersed signal for the major satellites in *Ago1*_KO mESCs, indicating that the overall structures of chromocenters is preserved in *Ago1*_KO mESCs (Fig S3D).

In summary, we observed that upon *Ago1* depletion, major satellite transcripts are up-regulated in mESCs and that their expression can be partially rescued by the re-expression of AGO1. Furthermore, this phenotype was not accompanied by the loss of chromocenters structure, as they could still form normally in *Ago1*_KO mESCs.

## MiRNAs are involved in the regulation of major satellites in mESCs

We then attempted to identify the molecular mechanism causing the up-regulation of major satellite transcripts in *Ago1*_KO mESCs. The AGO proteins are best known for their role in post-transcriptional silencing via miRNAs (Meister, 2013). To investigate the role of miRNAs in constitutive heterochromatin distribution in mESCs, we performed IF of H3K9me3 and HP1α in *Drosha*_KO mESCs (Cirera-Salinas et al, 2017), generated in the same background than our *Ago1*_KO mESCs. Interestingly, we observed a strong redistribution of H3K9me3 and HP1α away from the pericentromeric regions in *Drosha*_KO compared with WT mESCs, suggesting a role for miRNAs in the regulation of pericentromeric regions in mESCs (Fig 4A and B). In addition, we observed an up-regulation of HP1α in *Drosha*_KO mESCs (Fig S4A), suggesting that HP1α itself might be directly regulated by miRNAs.

MiRNA gene regulation is usually taking place in the cytoplasm of the cells (Bartel, 2018). Nevertheless, AGO2 was previously shown to shuttle into the nucleus (Nishi et al, 2013) and be enriched in mESC nucleus (Sarshad et al, 2018). Accordingly, we sought to assess the subcellular distribution of AGO1 in mESCs. Using well-established biochemical assays (Gagnon et al, 2014b), we performed cytoplasmic/nucleoplasmic/chromatin fractionation of WT mESCs and analyzed the abundance of AGO1 in these three fractions by WB (Fig 4C). We observed that the most of the AGO1 localized to the cytoplasm, whereas only around 10–15% of AGO1 is present in the nuclear fraction and even less than 3% is found in the chromatin fraction (Fig 4C). Cross-contamination was controlled for by using specific subcellular markers to validate the purity of the different fractions (Fig 4C). These results led us to hypothesize that, like AGO2, AGO1 loaded with miRNA is able to shuttle in the nucleus of mESCs and might target specific transcripts in the nucleus. We therefore attempted to identify whether miRNAs target major satellite transcripts in mESCs. Most major satellite annotations, as obtained from RepeatMasker and Dfam (Smit et al, 2013; Bao et al, 2015; Storer et al, 2021), were not properly mapped to any of the chromosomes and therefore annotated in unmapped genomic contigs (Fig S4B). In accordance with reports of major satellite sequences being several kilobases long, we selected those annotations that mapped to regions of at least 20 kbps of length, and focused on major satellite regions that mapped to chromosome X, 9 and the contigs JH583204.1 and GL456383.1. We searched for miRNAs with high confidence seed matches within these sequences and identified three miRNAs from the miR-30 family

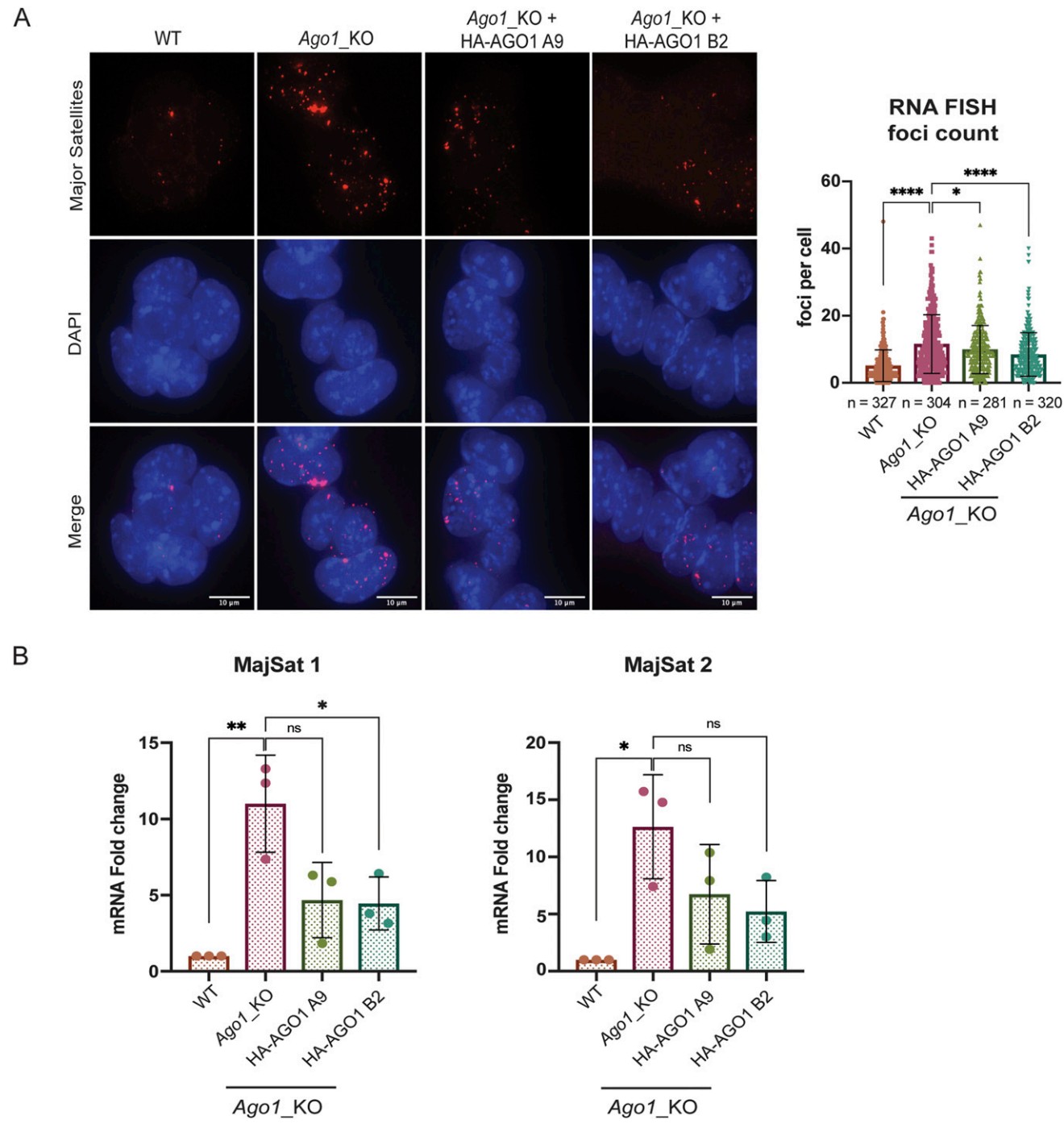

**Figure 3. Major satellite transcripts are up-regulated in *Ago1*_KO mouse embryonic stem Cells (mESCs).**
**(A)** Left: representative images of major satellite RNA FISH in WT, *Ago1*_KO, and two HA-AGO1 single clones, A9 and B2. Scale bar = 10 μm. Right: quantification of major satellite RNA FISH foci count in WT and *Ago1*_KO mESCs two HA-AGO1 single clones, A9 and B2. The graph shows the mean distribution with standard deviations. **** = *P*-value < 0.0001 and **** = *P*-value < 0.05 unpaired *t* test for n = 3 independent experiments. **(B)** RT-qPCR results for major satellite primer set 1 and 2 (Table S1) in WT and *Ago1*_KO mESCs and two HA-AGO1 single clones, A9 and B2. * = *P*-value < 0.05 and ** = *P*-value < 0.01, ns, not significant, unpaired *t* test for n = 3 independent experiments.

having a high number (more than 500) of 8mer binding sites (BS) within major satellite sequences (Fig 4D). In addition, two other miRNAs miR-139-5p and mR-6989-3p showed significant predicted BS. However, both these had much fewer binding sites compared with the miR-30-3p family (Fig 4D). Whereas, miR-139-5p has around 200 BS for the annotated region on the X chromosome, it has basically no BS for the other three regions. Also, miR-6989-3p has only around 100 BS in two contigs and even less in the others (Fig 4D).

The miR-30 family is composed of six pre-miRNAs (miR-30a, miR-30b, miR-30c-1, miR-30c-2, miR-30d, and miR-30e) located on three

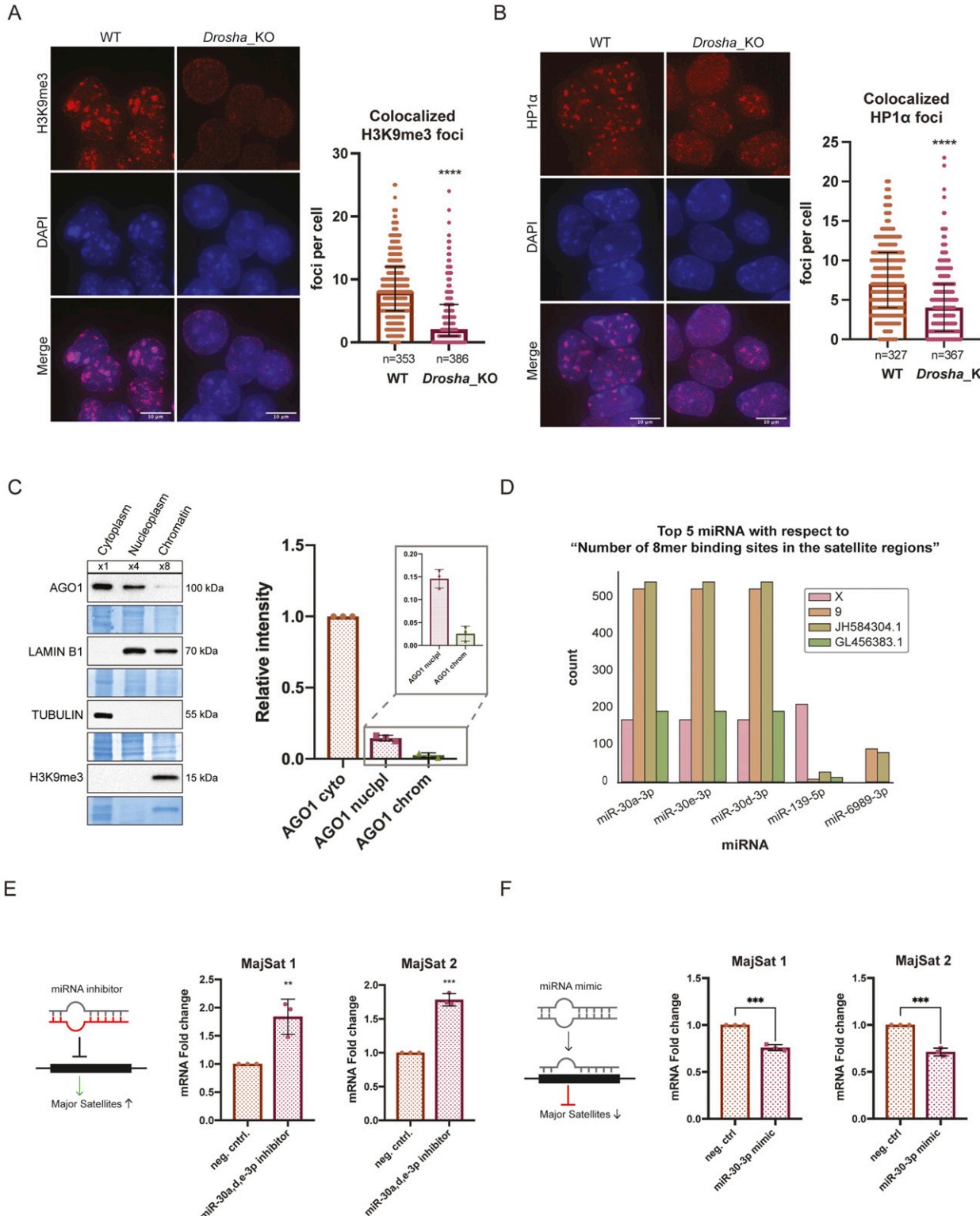

**Figure 4. miRNAs are involved in the regulation of major satellite transcripts.**
**(A)** Left: representative IF images of H3K9me3 in WT and *Drosha*_KO mouse embryonic stem cells (mESCs). Scale bar = 10 μm. Right: quantification of foci count for H3K9me3 that colocalizes with DAPI regions in WT and *Drosha*_KO mESCs. The graph shows the median distribution with the interquartile range. **** = *P*-value < 0.0001, Mann–Whitney test for n = 3 independent experiments. **(B)** Left: representative IF images of HP1α in WT and *Drosha*_KO mESCs. Right: quantification of foci count for HP1α that colocalizes with DAPI regions in WT and *Drosha*_KO mESCs. The graph shows the median distribution with the interquartile range. **** = *P*-value < 0.0001, Mann–Whitney test for n = 3 independent experiments. **(C)** Representative Western blots for the fractionation of WT mESCs to visualize AGO1 subcellular localization and

different chromosomes, which are all expressed in WT mESCs (Fig S4C and D and Table S2). Whereas all the mature miR-30-5p share the same seed sequence, only miR-30a-3p, miR-30d-3p, and miR-30e-3p have identical seeds, which match the major satellite sequences (Figs 4D and S4E). In addition, by re-analyzing our published AGO1 RNA immunoprecipitation and sequencing (RIP-seq) data (Ngondo et al, 2018), we identified that these three miRNAs are preferentially loaded into AGO1 (Fig S4F and Table S2). To investigate a possible regulation of major satellite transcripts by miR-30a-3p, miR-30d-3p, and miR-30e-3p, we used miRNA inhibitors against the three miRNAs in WT mESCs. We transfected WT mESCs either with a negative control inhibitor or with a pool of miR-30a-3p, miR-30d-3p, and miR-30e-3p inhibitors. We monitored the major satellite transcripts level 36 h after transfection by RT-qPCR and identified an increase in around twofold upon transfection with the miR-30a-3p, miR-30d-3p, and miR-30e-3p inhibitors compared with the negative control (Fig 4E). Similarly, transfecting WT mESCs with a miR-30-3p mimic significantly decreased major satellite transcript levels (Fig 4F). Taken together, these results indicate a role for the miR-30a-3p, miR-30d-3p, and miR-30e-3p in fine-tuning major satellite transcript levels.

## Discussion

Since the discovery that AGO proteins can localize to the nucleus in mammalian cells, numerous studies have attempted to describe their nuclear functions (Meister, 2013). Although in human cells, nuclear AGO's have been linked to functions in transcriptional gene regulation, splicing, chromatin organization, and double-strand break repair (Janowski et al, 2006; Kim et al, 2006; Li et al, 2006; Ameyar-Zazoua et al, 2012; Hu et al, 2012; Huang et al, 2013; Alló et al, 2014; Cho et al, 2014; Gao et al, 2014; Agirre et al, 2015; Portnoy et al, 2016; Wang & Goldstein, 2016; Shuaib et al, 2019), little is known about their role during early embryonic development.

In this study, we aimed to assess a possible role for AGO1 in the distribution of constitutive heterochromatin in mESCs. AGO1 had previously been reported to interact with RNA Polymerase II in human cells, where AGO1 was linked to chromatin and active promoters (Huang et al, 2013; Alló et al, 2014; Shuaib et al, 2019). We therefore decided to conduct a genetic approach by depleting Ago1 from WT mESCs (Fig S1A and B) and assessed by immunofluorescence the localization of the repressive histone mark H3K9me3 and the heterochromatin protein HP1α in the mutant cell lines compared with WT mESCs (Fig 1). Surprisingly, we observed a redistribution of both H3K9me3 and HP1α away from pericentromeric regions in Ago1_KO mESCs (Fig 1). The redistribution of H3K9me3 and HP1α was found to be specific to the loss of AGO1, as reintroducing AGO1 could rescue the phenotype (Fig 2). We questioned whether major satellites residing within pericentromeric regions

are up-regulated at the transcript level in Ago1_KO mESCs. Indeed, we observed an increase in pericentromeric major satellite transcripts in Ago1_KO mESCs by RT-qPCR and RNA FISH, which could again be rescued by reintroducing AGO1 into Ago1_KO mESCs (Fig 3). This increase was not caused by a change in the number of chromocenters, as was confirmed by DNA FISH (Fig S3C and D). Finally, we wondered whether AGO1 could regulate pericentromeric transcripts by a miRNA-mediated mechanism and observed similar delocalization of both H3K9me3 and HP1α away from pericentromeric regions also in Drosha_KO mESCs, suggesting a role for miRNAs (Fig 4A and B). Using computational analysis, we identified that miR-30a-3p, miR-30d-3p, and miR-30e-3p might target major satellite transcripts and that manipulating the amount of these miRNAs in WT mESCs using inhibitors or mimics inversely regulates major satellite transcript levels to some extent (Fig 4D–F). In addition, by analyzing the subcellular distribution of the AGO1, we found that a small fraction (10–15%) localized to the nucleus, leading us to the hypothesis that AGO1 loaded with miR-30-3p might directly regulate major satellite transcripts. Of note, the regulation observed here using mimics or inhibitors (Fig 4E and F) was lower (twofold) than the one observed comparing WT and Ago1_KO mESCs (10-fold) (Fig 3B). We also observed an up-regulation of HP1α at protein level in Ago1_KO (Fig S1F) and Drosha_KO mESCs (Fig S4A), leading us to propose that HP1α might also be directly regulated by miRNAs in mESCs (as also suggested by several miRNA binding sites in its 3′UTR and AGO2-binding sites [Schäfer et al, 2021 Preprint]). It would be interesting to further investigate this direct regulation of HP1α by miRNAs in a follow-up study.

An involvement of AGO proteins loaded with small RNAs in the regulation of pericentromeric regions has previously been reported in Schizosaccharomyces pombe, where Ago1 loaded with siRNAs is guided to pericentromeres (Verdel et al, 2004). Ago1 together with Tas3 and Chp1 forms the RNA-induced transcriptional silencing complex (RITS). The RITS is guided to centromeric repeats by siRNAs, which are derived from this region. Targeting the RITS complex to centromeric repeats is needed for the localization of the HP1α homolog Swi6 and the nucleation of heterochromatin H3K9me at these sites (Motamedi et al, 2004; Verdel et al, 2004; Bühler et al, 2006; Goto & Nakayama, 2012). However, our findings differ from the ones in yeast as we did not identify any small RNAs, derived from pericentromeric regions to be loaded in AGO1. However, there have been reports suggesting the presence of small RNAs from pericentromeric regions in mammalian cells (Kanellopoulou et al, 2005; Hsieh et al, 2011), we found that AGO1 in mESCs is probably guided to major satellite transcripts by specific miRNAs, miR-30a-3p, miR-30d-3p, and miR-30e-3p.

Importantly, even though we identified a decrease in H3K9me3 at pericentromeric regions upon the depletion of Ago1 in mESCs, this was accompanied by only a small impact on the global

quantification of n = 3 independent experiments. LAMIN B1 (nucleoplasm and chromatin), TUBULIN (cytoplasm), and H3K9me3 (chromatin) were used as subcellular markers. **(D)** Representation of the top 5 miRNAs with 8mer binding sites targeting major satellite sequences located on chromosomes X, 9 and the genomic contigs JH584304.1 and GL456383.1. **(E)** RT-qPCR results for major satellite primer sets 1 and 2 (Table S1) in WT mESCs transfected with a negative control inhibitor and a pool of inhibitors against miR-30a, d, e-3p. ** = $P$-value < 0.01 and *** = $P$-value < 0.001, unpaired $t$ test for n = 3 independent experiments. **(F)** RT-qPCR results for major satellite primer sets 1 and 2 (Table S1) in WT mESCs transfected with a negative control mimic and a miR-30-3p mimic. *** = $P$-value < 0.001, unpaired $t$ test for n = 3 independent experiments.

transcriptome and none regarding viability of *Ago1*_KO mESCs (Ngondo et al, 2018; Mueller et al, 2021 *Preprint*). Ngondo et al (2018), have reported that the depletion of *Ago1* does not affect the cell cycle nor their potential to differentiate (Van Stry et al, 2012; Ngondo et al, 2018). It appears that the loss of HP1α and H3K9me3 disturbs the environment more locally without affecting overall cell viability. Although we do not currently know how mESCs cope with this loss, it is possible that the plasticity of stem cells or the reestablishment of heterochromatin at pericentromeric regions upon differentiation may be required for survival. As several studies in human and cancer cells have already described a nuclear role for the AGO proteins, especially also for AGO1 (Janowski et al, 2006; Kim et al, 2006; Li et al, 2006; Ameyar-Zazoua et al, 2012; Hu et al, 2012; Huang et al, 2013; Alló et al, 2014; Cho et al, 2014; Agirre et al, 2015; Portnoy et al, 2016; Shuaib et al, 2019), it will be interesting to study whether the decrease of H3K9me3 at pericentromeric regions also occurs in these cell types. Interestingly, the up-regulation of major satellite transcripts in several cancer lines has already been described (Hall et al, 2012); however, we do not know whether this might be linked to a nuclear AGO1 function.

There are still open questions and further experiments required to identify the complete underlying molecular mechanism. How AGO1 regulates heterochromatin at pericentromeric regions and major satellite transcripts remains an open question. Our attempts to localize AGO1 at pericentromeric regions using IF or ChIP-qPCR approaches, using specific antibodies or AGO1-tagged cell lines, remained unsuccessful (data not shown). These negative results might come from the low amount of AGO1 in the nucleus (Fig 4C). New approaches or technical development are therefore required to precisely address AGO1 localization in the nucleus or chromatin in mESCs. Furthermore, to dissect a direct role of AGO1, it might be helpful in the future to assess whether AGO1 RNA binding activity is required for direct major satellite targeting via miRNAs.

In conclusion, our study reports a novel role for AGO1 in the nucleus of mESCs and we believe that these observations might help to motivate future research on the AGO proteins in early embryonic development.

## Materials and Methods

### Mouse ESC lines

WT E14 (129/Ola background), *Ago1*_KO, AGO1 complemented *Ago1*_KO, and *Drosha*_KO (Cirera-Salinas et al, 2017) mESCs were cultured in DMEM (Sigma-Aldrich), supplemented with 15% FBS (Life Technologies), 100 U/ml LIF (Millipore), 0.1 mM 2-β-mercaptoethanol (Life Technologies), and 1% penicillin/streptomycin (Sigma-Aldrich). MESCs were cultured on 0.2% gelatin-coated culture flasks in the absence of feeder cells. The culture medium was changed daily and all cells were grown at 37°C in 8% $CO_2$.

### CRISPR/Cas9–mediated gene knockout

The generation of the *Ago1*_KO1 cell line was previously published by (Ngondo et al, 2018). The *Ago1*_KO2 cell line was generated using

a paired CRISPR/Cas9 approach, as described by Wettstein et al (2016). E14 mESCs were transfected with lipofectamine 2000 (Invitrogen) and the pX458-sgRNA_*Ago1*_5/6 (#172470, #172471; Addgene). After 48 h, GFP-positive cells were single sorted into 96-well plates (TPP). To confirm the deletion, genotyping at DNA level was performed, with the primers PS_Ago1_FW/RW_1 listed in Table S1. MESC clones were then amplified and the absence of AGO1 protein and RNA was additionally verified by Western blotting and RT-qPCR, respectively.

### Generation of *Ago1*_KO complemented cell lines

For the rescue experiments, the AGO1 complemented *Ago1*_KO2 cells were obtained by stably transfecting the pMSCV_PIG_3xHA-AGO1 plasmid (#170916; Addgene) with lipofectamine 3000 (Invitrogen). Cells were grown for 1 wk under puromycin selection and then sorted by FACS to select only GFP expressing cells. We sorted two mixed population into separate dishes of around 10,000–20,000 cells. The mixed populations were expanded and the expression of HA-AGO1 was tested by Western blot and Immunofluorescence (Figs 2A and S2A and B). For single clone generation, single cells expressing GFP were sorted into a 96-well plate and expanded. The expression of HA-AGO1 was tested by Western blot (Fig S2C and D).

### Cytoplasmic/nucleoplasmic/chromatin fractionation

Cytoplasmic/nucleoplasmic/chromatin fractionation was performed after the protocol of Gagnon et al (2014b). Cells were grown to near confluency in two 75 cm$^2$ (T75) flasks (TPP). 10 millions of WT mESCs were used. Freshly harvested cells were incubated for 10 min in ice-cold Hypotonic lysis buffer complemented with EDTA-free protease inhibitor cocktail (Roche) and Phosphatase inhibitor cocktail (Roche). After centrifugation (800*g* for 8 min at 4°C) the cytoplasmic fraction was transferred to a new tube containing 5M NaCl. Pellets were washed four times with Hypotonic lysis buffer (200*g* for 2 min). After the last wash ice-cold modified Wuarin-Schipler buffer (MWS) (10 mM Tris–HCl, [pH 7.0], 4 mM EDTA, 0.3 M NaCl, 1 M urea, and 1% [vol/vol] IGEPAL-C630), complemented with EDTA-free protease inhibitor cocktail and Phosphatase inhibitor cocktail, was added and after vortexing, incubated for 15 min on ice. After centrifugation (1,000*g* for 5 min at 4°C), the nucleoplasmic fraction was transferred to a new tube. The chromatin pellet was washed twice with MWS buffer, vortexed, incubated on ice for 5 min, and centrifuged at 500*g* for 3 min at 4°C. Ice-cold NLB was added to the chromatin pellet, which was sonicated twice at 20% for 15 s with 2 min incubations on ice in between. The three fractions were centrifuged for 15 min at 18,000*g* and the supernatant was transferred to a new tube.

The fractions were then analyzed by Western blot. To ensure proper representation of all the fractions, more of the nuclear (×4) and the chromatin (×8) fraction were loaded (Fig 4C).

Analysis of the Western blot signal was performed using ImageLab (Bio-Rad Laboratories). The intensity of the bands was calculated relative to the WT band. The intensities of the nuclear and chromatin fractions were adjusted according to

the additional loading and the fact that they were resuspended in half the amount of buffer compared with the cytoplasm.

## Western blot analysis

Total cellular proteins were extracted using RIPA lysing buffer (50 mM Tris–HCL, pH 8.0, 150 mM NaCl, 1% IGEPAL-CA630, 0.5% sodium deoxycholate, and 0.5% sodium dodecyl sulfate supplemented with EDTA-free protease inhibitor cocktail [Roche]). Protein concentration was determined by a Bradford assay (Bio-Rad Laboratories). Proteins were separated on an SDS–PAGE gel and transferred to a polyvinylidene difluoride membrane (Sigma-Aldrich). Membranes were blocked for at least 30 min in blocking solution (5% milk in 1X TBS-T: TBS, pH 7.6: 50 mM Tris–HCL, 150 mM NaCl, and 0.1% Tween 20) and incubated overnight with primary antibodies diluted in blocking solution at 4°C. Primary antibodies used were: HP1$\alpha$ (#2616, 1:2,000; CST), AGO1 (#5053, 1:2,000; CST), LAMIN B1 (ab16048, 1:10,000; Abcam), TUBULIN (T6199, 1:10,000; Sigma-Aldrich), H3K9me3 (ab8898, 1:2,000; Abcam), and HA (3F10, 1:2,000; Roche).

After washing three times in 1× TBS-T for 10 min, membranes were incubated with the secondary antibody for 1 h at room temperature (rabbit-IgG HRP-linked 1:10,000; Cell Signaling Technology [#7074], mouse-IgG HRP-linked 1:10,000; Cell Signaling Technology [#7076], rat-IgG HRP-linked 1:10,000 [#7077]). After incubation, membranes were washed again three times 10 min in 1X TBS-T and developed using the Clarify Western ECL substrate kit (Bio-Rad) or SuperSignal West Femto (Thermo Fisher Scientific). Membranes were imaged using the ChemiDoc MP imaging system (Bio-Rad Laboratories).

Analysis of the Western blot signal was performed using ImageLab (Bio-Rad Laboratories). Coomassie or TUBULIN was used as normalizer. Intensities of the bands were calculated relative to the WT band.

## Immunofluorescence and analysis

Approximately 100,000 cells were plated the night before into six-well plates (TPP), containing coverslips coated with fibronectin (1:100 in 1× PBS; Merck). The next day, cells were washed once with 1× PBS.

For the H3K9me3 and HA staining, cells were fixed with ice-cold Methanol for 10 min at −20°C. After fixation they were washed three times with 1× PBS and blocked for 20 min in blocking solution (1% BSA in 1X PBS-Tween 20 [0.1%]).

For the HP1$\alpha$ staining, a nuclear pre-extraction was performed. Cells were washed once with ice-cold 1× PBS for 3 min on ice and then incubated in CSK buffer (0.1% Triton X-100, 10 mM PIPES, 100 mM NaCl, 3 mM MgCl$_2$, and 300 mM Sucrose) for 3 min, also on ice. Afterwards cells were washed once with 1× PBS and fixed with 3.7% formaldehyde (Sigma-Aldrich) for 10 min at room temperature. After fixation, cells were washed twice with 1× PBS for 5 min at room temperature and then permeabilized with CSK buffer (same as above) for 4 min on ice. After two additional wash steps with 1× PBS at room temperature, cells were blocked in blocking solution (1% BSA in 1× PBS-Tween 20 [0.1%]) for 20 min at room temperature.

After blocking, cells were incubated with the primary antibodies diluted in blocking solution (H3K9me3: ab8898, 1:500, HP1$\alpha$: #2616, 1:200; CST, HA: 3F10, 1:250; Roche) for 1 h at room temperature. Coverslips were washed three times for 5 min at room temperature with 1× PBS-Tween 20 (0.1%). Then, cells were incubated with secondary antibodies diluted in blocking solution (1:2,000; Invitrogen) for 1 h at room temperature in the dark. Again, coverslips were washed three times for 5 min at room temperature with 1× PBS-Tween 20 (0.1%) and once with 1× PBS. Counterstain with DAPI (0.1 $\mu$l/ml) in 1× PBS was performed for 4 min at room temperature. Cells were washed once with 1× PBS and mounted on microscopy slides on a drop of antifade medium (Vectashield; Vector Laboratories). Slides were imaged on a DeltaVision Multiplexed system with an Olympus IX71 inverse microscope equipped with a 60× 1.4NA DIC Oil PlanApoN objective and a pco.edge 5.5 camera, provided by the ScopeM facility of ETH.

For image analysis, deconvolved images were processed with Fiji (Schindelin et al, 2012). A Z-projection of the Max intensity has been performed for each image and used for further analysis. Foci count and intensity analysis was performed on the Z-projected images, with the help of CellProfiler (Mcquin et al, 2018). In CellProfiler, nuclei were identified by using the IdentifyPrimaryObjects module and the Otsu thresholding method. Nuclei were edited manually and the DAPI and H3K9me3 was enhanced with the EnhanceOrSupressFeatures module to detect speckles. Foci were identified by using IdentifyPrimaryObject. For the DAPI foci, the RobustBackground was used as a thresholding method and the threshold strategy was set to Global. Typical diameter of objects, in pixel units was set to 5–35. For the H3K9me3 foci the same thresholding method and strategy was used, but the typical diameter of objects, in pixel units was set to 7–35. Foci were related to the edited nuclei and the H3K9me3 foci were related to the DAPI foci. The MeasureObjectIntensity module was used to measure object intensity and foci count and results were exported to a csv file. Overlaid objects were saved as png.

## RNA extraction and quantitative RT-qPCR analysis

RNA extraction and RT PCR analysis has been performed as previously described by Bodak and Ciaudo (2016). Briefly, total RNA from mESC pellets was extracted using Trizol (Life Technologies) according to standard protocols. RNA quality was checked, by running 1 $\mu$g on a 1% agarose gel (Sigma-Aldrich).

For qPCR on major satellite transcripts, 20 $\mu$g of RNA was treated twice with 1U of DNaseI (QIAGEN) per $\mu$g of RNA. RiboLock was added to reduce RNA degradation. DNase-treated RNA was purified using Direct-zol RNA mini prep kit (Zymo Research). Reverse transcription and qPCR were performed as described above. Primers are listed in the Table S1.

# RNA FISH

The plasmid pCR4 Maj9-2 (a kind gift from the Almouzni laboratory, originally from Lehnertz et al [2003]) was used to generate the RNA FISH probe by nick translation (Abott). In brief, 2 $\mu$g of plasmid, 3 $\mu$l of nick translation enzyme, 2.5 $\mu$l 0.2 mM red-dUTP, 5 $\mu$l 0.1 mM dTTP,

10 µl 0.1 mM dNTP mix, and 5 µl 10× nick translation buffer were incubated for 15 h at 15°C. Nick translation efficiency was checked for by running 3 µl probe on a 1% agarose gel (Sigma-Aldrich). The rest of the probe was cleaned-up with the Zymoclean Gel DNA Recovery Kit (Zymo Research). The probe was dried down to 5 µl using a speed vac and then resuspended in hybridization solution (50 µl Deionized Formamide, 10 µl 20X SSC, 2 µl 100 mg/ml BSA, 20 µl 50% Dextran Sulfate, 3 µl Salmon Sperm, 10 µl RVC). Before use, the probe was diluted 1:2 in hybridization solution.

Approximately 150,000 cells were plated the night before into six-well plates, containing coverslips coated with fibronectin (1:100 in 1X PBS; Merck). A nuclear pre-extraction was performed. Cells were washed once with ice-cold 1× PBS for 3 min on ice and then incubated in CSK buffer (0.1% Triton X-100, 10 mM PIPES, 100 mM NaCl, 3 mM $MgCl_2$, and 300 mM sucrose) for 3 min, also on ice. Afterwards, the cells were washed once with 1× PBS and fixed with 3.7% formaldehyde (Sigma-Aldrich) for 10 min at room temperature. After fixation, cells were washed twice with 1× PBS for 5 min at room temperature and then permeabilized with CSK buffer (same as above) for 4 min on ice. After two additional wash steps with 1× PBS at room temperature, cells were blocked in blocking solution (1% BSA in 1X PBS and RVC [1 mM]) for 30 min at room temperature. Then coverslips were washed once with 2× SSC and dehydrated with ethanol (70% EtOH for 3 min, 90% EtOH for 3 min, and 100% EtOH for 3 min). The probe was denatured for 5 min at 76°C. 10 µl of the denatured probe (diluted 1:2 in hybridization buffer) was spotted on a baked slide. The coverslips were air-dried and placed on the spotted probe. The coverslips were sealed with rubber cement and then incubated overnight at 37°C in a humid chamber. The next day, coverslips were washed twice with 50% formamide/2× SSC for 5 min at 37°C and then once for 5 min with 2× SSC at room temperature. Counterstain with DAPI (0.1 µl/ml) in 2× SSC was performed for 4 min at room temperature. Coverslips were washed again once in 2× SSC and once 1× PBS and then mounted on microscopy slides on a drop of antifade medium (Vectashield; Vector Laboratories). Slides were image on a DeltaVision Multiplexed system provided by the ScopeM facility of ETH as above.

For image analysis, deconvolved images were processed with Fiji (Schindelin et al, 2012). A Z-projection of the Max intensity has been performed for each image and used for further analysis. RNA Foci were counted by eye. All Z-projected images of one replicate were opened using the Fiji software and were set to the same intensity.

## DNA FISH

DNA FISH was performed exactly as described for the RNA FISH. The only differences, are that no RVC was used in the blocking solution and the samples were denatured by incubating the slides at 76°C for 5 min once the coverslips were placed on the spotted probe and sealed with rubber cement.

### ChIP and ChIP-qPCR analysis

Four million cells were plated the night before into a gelatin-coated 60.1 cm² (B10) dish (TPP). For each condition, two B10 dishes were prepared in parallel. Cells were cross-linked with 1% formaldehyde in DMEM for 10 min at room temperature. The reaction was quenched with glycine (125 mM; PanReac Applichem) for 5 min at room temperature. Cells were washed once with ice-cold 1× PBS and then swelling buffer (5 mM Hepes, pH 8, 85 mM KCl, 0.5% IGEPAL-CA630, and protease inhibitor cocktail [Roche]) was added to the cells. Cells were scraped and transferred to a 15 ml falcon (Greiner), where they were incubated for 15 min on ice. Cells were centrifuged 5 min at 250g (Eppendorf Centrifuge 5810R; Rotor A-4-81 [for Falcon tubes]) at 4°C, to pellet the nuclei. Afterwards, nuclei were washed again with swelling buffer followed by another centrifugation (250g at 4°C for 5 min). The nuclei pellet was lysed in 400 µl RIPA buffer 1% SDS (1× PBS, 1% IGEPAL-C630, 0.5% sodium deoxycholate, 1% sodium dodecyl sulfate, and protease inhibitor cocktail [Roche]) and incubated on ice for 10 min. The lysates were sonicated on a Bioruptor (Diagenode) for 30 min, 30 s on and 30 s off cycles at 4°C. Lysates were centrifuged at max speed for 15 min at 4°C. The supernatant was retrieved into a new 2 ml Eppendorf tube and diluted 10 times with RIPA buffer 0% SDS (1× PBS, 1% IGEPAL-C630, 0.5% sodium deoxycholate, and protease inhibitor cocktail [Roche]) to obtain a concentration of 0.1% sodium dodecyl sulfate. 10% of chromatin was taken away for Input calculation, the rest of the chromatin was snap-frozen and stored at −80°C. Input DNA was treated with 10 µg RNase A for 1 h at 37°C followed by a proteinase K treatment (40 µg) for 1–2 h. DNA was extracted with phenol/chloroform (Sigma-Aldrich) and concentration was measured and used to calculate the total amount of chromatin in each sample.

For the pull-down, 20 µg of chromatin was precleared with 10 µl of Dynabeads protein G (Thermo Fisher Scientific), previously washed three times with RIPA 0.1% SDS (1× PBS, 1% IGEPAL-C630, 0.5% sodium deoxycholate, 0.1% sodium dodecyl sulfate, and protease inhibitor cocktail [Roche]), for 2 h on the wheel at 4°C. 1/10 of the precleared chromatin was taken away and stored temporarily at −20°C, this was later used as the Input. The rest of the precleared chromatin was transferred into a new 1.5 ml Eppendorf tube and incubated with 2 µg of antibody for each condition overnight at 4°C (H3K9me3: ab8898, rabbit-IgG: NI01). 10 µl of Dynabeads protein G (Thermo Fisher Scientific) was added to the chromatin-antibody complexes and incubated 4 h on the wheel at 4°C. Samples were placed on the magnetic rack and the supernatant was discarded. Samples were washed twice with wash buffer 1 (16.7 mM Tris–HCL, pH 8, 0.167 M NaCl, 0.1% SDS, and 1% Triton X-100) for 5 min rotating at room temperature. Then they were washed once with wash buffer 2 (16.7 mM Tris–HCl, pH 8, 0.5M NaCl, 0.1% SDS, and 1% Triton X-100) for 5 min rotating at room temperature and twice in LiCl wash buffer (0.25 M LiCl, 0.5% sodium deoxycholate, 1 mM EDTA, pH 8, 10 mM Tris–HCl, pH 8, and 0.5% IGEPAL-CA630) for 5 min rotating at room temperature. Finally, the samples were washed twice in TE buffer (10 mM Tris–HCl, pH 8, and 5 mM EDTA, pH 8) for 5 min rotating at room temperature. Samples were incubated in 300 µl elution buffer (1% SDS, 100 mM NaHCO₃) for 30 min at 37°C shaking at 900 rpm (Eppendorf ThermoMixer C). Samples were placed on a magnetic rack and the supernatant transferred into a new Eppendorf tube containing 38.5 µl Proteinase K mix (15 µl 1M Tris–HCl, pH 8, 15 µl 5M NaCl, 7.5 µl 0.5M EDTA, pH 8, and 1 µl Proteinase K [20 mg/ml]). Also 300 µl elution buffer and 38.5 µl Proteinase K mix was added to the Inputs. Pull-downs and Inputs were incubated at 50°C for 3 h shaking at 1,100 rpm (Eppendorf

ThermoMixer C) and then at 65°C overnight. DNA was treated with 10 μg of RNase A for 45 min at 37°C and extracted with a phenol/chloroform extraction, followed by an ethanol precipitation.

ChIPed and Input DNA was diluted 1:10 before the qPCR for the control Primers (Dazl, MusD and the intergenic region) and 1:50 before the qPCR for the major satellites. The qPCR was performed with the KAPA SYBR Fast qPCR Kit (Kapa Biosystems) and analyzed on a LightCycler 480 (Roche). The enrichment was calculated with the $2^{-\Delta\Delta CT}$ method over input. Control regions (Dazl, MusD [Karimi et al, 2011]) were represented as enrichment over the intergenic region (Ngondo et al, 2018). The enrichment for the major satellites was represented as the enrichment compared with WT. Primers are listed in Table S1.

### Major satellite computational analysis and binding site identification

RepeatMasker annotations were obtained from UCSC for the mm10 mouse reference genome (http://hgdownload.soe.ucsc.edu/goldenPath/mm10/database/rmskOutCurrent.txt.gz) and filtered for major satellites (GSAT_MM) (Bao et al, 2015; Smit et al, 2013). Most regions annotated as major satellites were rather short (<1,000 bps) and we only considered four regions, where the genome sequence contained > 20 Kbps long major satellite regions. One of them was mapped into the X chromosome and another one to chromosome 9. The other two fell into genomic contigs that could not be assigned to any chromosome (JH584304.1 and GL456383.1). For these four annotated regions, 8mer-binding sites were scanned and counted for each mESC-expressed miRNA (Table S2). Small RNA-seq from WT mESCs has been previously analyzed in Schäfer et al (2021) Preprint and RIP-small RNA-seq in Ngondo et al (2018).

### miRNA inhibitor and mimic transfection

Approximately 300,000–400,000 WT mESCs were plated the night before into a gelatin-coated 60.1 cm² (B10) dish (TPP). The next day, for the miRNA inhibitor transfection, WT mESCs were either transfected with RNAiMax (Invitrogen) and 30 nM of a negative control inhibitor (#4464074; Ambion) or with RNAiMax (Invitrogen) and a mix of miR-30a-3p, miR-30d-3p, and miR-30e-3p inhibitors, 10 nM for each inhibitor (#4464084; Ambion). For the miRNA mimic transfection, WT mESCs were either transfected with RNAiMax (Invitrogen) and 30 nM of a negative control mimic (CN-001000-01-05; Dharmacon) or with RNAiMax (Invitrogen) and 30 nM of miR-30e-3p mimics (C-310467-07-0002; Dharmacon). 36 h later, the cell pellet was collected. Briefly, cells were washed once with 1× PBS (Life Technologies), then trypsinized with 0.05% Trypsin–EDTA (Life Technologies) for 5 min at 37°C. Trypsinization was stopped, by adding medium and spinning the cells down for 5 min at 182g. The cell pellet was washed once in 1× PBS (Life Technologies), spun down 5 min at 182g and then stored at −80°C.

### Quantification and statistical analysis

See Methods Details for details on quantification and statistical analysis. In general, statistical analysis was performed using PRIMS 8 as indicated in the figure legends.

## Data Availability

Small RNA-seq (GSE80415) and RIP-small RNA-seq (GSE80454) (Ngondo et al, 2018) used in this study have been deposited in the Gene Expression Omnibus repository, analyzed results are provided in Table S2. References of images: FACS machine in Fig S3: https://fluorofinder.com/cytometer-facsaria/. 96-well plate: https://commons.wikimedia.org/wiki/File:96-Well_plate.svg.

## Supplementary Information

## Acknowledgements

We would like to thank the members of the Ciaudo Lab, Dr. Tobias Beyer, and Prof. Madhav Jagannathan (Swiss Federal Institute of Technology [ETH]) for fruitful discussions and the critical reading of this manuscript. We would also like to thank the Santoro lab for help with the ChIP protocol. This work was supported by the Swiss National Science Foundation (grants 31003A_173120 and 310030_196861) to C Ciaudo, R Arora and M Müller were supported by the National Center for Competence in Research (NCCR) RNA and disease. We are also thankful to the Scientific Centre for Optical and Electron Microscopy (ScopeM, ETH Zurich) for their support for imaging and the Flow Cytometry Core Facility of ETH for their help with FACS.

### Author Contributions

M Mueller: conceptualization, formal analysis, supervision, validation, investigation, visualization, methodology, and writing—original draft, review, and editing.
T Faeh: conceptualization, formal analysis, validation, investigation, visualization, methodology, and writing—original draft, review, and editing.
M Schaefer: data curation, software, visualization, methodology, and writing—review and editing.
V Hermes: formal analysis, investigation, and visualization.
J Luitz: formal analysis and investigation.
P Stalder: formal analysis and investigation.
R Arora: supervision, investigation, methodology, and writing—review and editing.
RP Ngondo: formal analysis, supervision, investigation, and writing—review and editing.
C Ciaudo: conceptualization, resources, formal analysis, supervision, funding acquisition, visualization, methodology, project administration, and writing—original draft, review, and editing.

### Conflict of Interest Statement

The authors declare that they have no conflict of interest.

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
