## [Reviewer comments · Life Science Alliance]

Life Science Alliance

AGO1 regulates pericentromeric regions in mouse embryonic stem cells

Madlen Mueller, Tara Faeh, Moritz Schaefer, Victoria Hermes, Janina Luitz, Patrick Stalder, Rajika Arora, Richard Patryk Ngondo, and Constance Ciaudo

DOI: <https://doi.org/10.26508/lsa.202101277>

Corresponding author(s): Constance Ciaudo, ETH Zurich

Review Timeline:

Submission Date:	2021-10-25
Editorial Decision:	2021-10-25
Revision Received:	2022-02-08
Editorial Decision:	2022-02-14
Revision Received:	2022-02-17
Accepted:	2022-02-17

Transaction Report:

Please note that the manuscript was reviewed at Review Commons and these reports were taken into account in the decision-making process at Life Science Alliance.

October 25, 2021

Re: Life Science Alliance manuscript #LSA-2021-01277-T

Prof. Constance Ciaudo
ETHZ
D-Biol
IMHS HPL G32.1
Otto-Stern-Weg 7
Zurich 8093
Switzerland

Dear Dr. Ciaudo,

Thank you for submitting your manuscript entitled "AGO1 regulates major satellite transcripts and H3K9me3 distribution at pericentromeric regions in mESCs" to Life Science Alliance. We invite you to re-submit the manuscript, revised according to your Revision Plan.

Thank you for this interesting contribution to Life Science Alliance. We are looking forward to receiving your revised manuscript.

Sincerely,

Eric Sawey, PhD
Executive Editor
Life Science Alliance
<http://www.lsa-journal.org>

B. MANUSCRIPT ORGANIZATION AND FORMATTING:

Dear Dr Sawey,

We now restructured our manuscript according to original reviewer requests and newly generated experiments. All changes have been highlighted in our revised manuscript. Please see below (in green) our point-by-point clarifications about these revisions.

1. HP1 α /Dnmt3A IFs in AGO1 complemented cell lines

Upon request by the reviewers, we plan to perform IF experiments in triplicates for HP1 α and Dnmt3A in the *Ago1_KO* +3x HA-AGO1 complemented mixed populations. These IFs will be then quantified as already done in Fig 2A and EV2E.

We have now performed the IFs in two independent *Ago1_KO* clones complemented with 3x HA-AGO1 (A9 and B2) and have quantified them. The HP1 α IF has been integrated in the figure 2. Single complemented clones were chosen since for IFs a nuclear pre-extraction was required for proper visualization of HP1 α that resulted in a rather weak signal for HA-AGO1 in the polyclonal population, making it difficult to choose cells where AGO1 was being re-expressed.

As explained previously, we faced some difficulties to validate our MS results by Co-IP followed by WB for AGO1 and HP1 α as well as DNMT3A. As requested by the original reviewers, we deleted the MS data from the paper and focus on the other observed phenotype.

2. Assess methylation status by IF (reviewer 2)

Reviewer 2 asked about the methylation patterns in *Ago1_KO* mESCs, due to our results on DNMT3A. Therefore, we plan to assess the methylation status by IF using a specific antibody in WT versus *Ago1_KO* in triplicates.

We have tried to assess the methylation pattern in our *Ago1_KO* mESCs by IF with an Anti-5-methylcytosine (MABE146, Merck) antibody. However, we could not detect any specific signal in WT mESCs. The signal seemed to be unspecific and cytoplasmic:

Figure 1. IF images in WT mESCs to stain for 5-methylcytosine (antibody used: MABE146, Merck).

In collaboration, we have also performed MS to detect 5meC in WT and *Ago1*_KO mESCs in duplicates. As a control we measured WT cells grown in 2i, which should have less methylation and a *Dnmt*_TKO cell line, which should have no methylation at all. From these preliminary results, we observed at least no decrease in methylation levels in *Ago1*_KO mESCs:

Figure 2. Graph showing the % of dC methylation in WT serum+LIF, WT 2i+LIF, *Ago1*_KO and *Dnmt*_TKO mESCs as measured by MS.

As we dropped the MS data from the manuscript removing the link with DNMT3A, we decided to also not include any methylation data in our manuscript.

3. Assess HP1 α upon miR-30-3p inhibitor transfection

Reviewer 3 ask us to show the effect of miR-30 inhibitors on HP1 α recruitment and H3K9me3 deposition at pericentromeric foci. We have already performed this for H3K9me3, but not yet for HP1 α . We plan to do this experiment, by transfecting WT mESCs with a negative control inhibitor and miR-30a, d, e-3p inhibitors followed by an HP1 α IF in triplicate.

We had previously tried to monitor the distribution of H3K9me3 upon miR-30-3p inhibitor transfections. However, as mentioned before, the transfection efficiency is rather low in mESCs (less than 50%). Combined with the absence of tools to identify cells that have taken up the inhibitors, it is not surprising that we only observed a mild and statistically not significant decrease of H3K9me3 upon transient miR-30-3p inhibitor transfection, as we consider the entire cell population and not just the transfected cells.

Figure 3. H3K9me3 IF in mir-30-3p inhibitor transfected mESCs. Depicted on the left side are IF images of WT mESCs transfected either with a negative control inhibitor or with miR-30-3p inhibitors. Depicted on the right side is the H3K9me3 foci quantification. Shown is the median distribution. pvalue = 0.1522, Mann-Whitney test. n=3 independent experiments.

We argue that it would be equally difficult to assess HP1 α levels upon miR-30-3p inhibitor transfection. Therefore, we decided to perform another experiment to investigate the importance of miRNAs in the distribution of H3K9me3 and HP1 α . In our revised manuscript, we now present (Fig 4) the quantification of H3K9me3 and HP1 α at pericentromeric regions after IF in WT versus *Drosha*_{KO} mESCs. DROSHA is an essential miRNA biogenesis factor. Similarly to *Ago1*_{KO} cells, we observed a significant decrease of H3K9me3 and HP1 α foci in *Drosha*_{KO} compared to WT mESCs, suggesting a role for miRNAs in the regulation of pericentromeric regions in mESCs.

1. We agreed with the comments of all three reviewers and have refocused our paper only on the AGO1 phenotype. All the changes have been already integrated in the revised manuscript and figures now submitted to LSA.

2. All three reviewers have asked us about the validations of the interactors identified by MS. We have now added these validations to our first supplemental figure (Revised Fig EV1C). We tried to co-IP ATRX with AGO1, but did not observe an interaction (data not shown). We also tried to co-IP HP1 α with AGO1, but we were also unsuccessful. Because we pulled down AGO1 with a rabbit antibody, we had to switch to another HP1 α antibody (goat anti- HP1 α , ab77256) that did not work well on our extracts. As HP1 α is 25kDA, which also corresponds to the lower IgG chain, we could not use the rabbit anti- HP1 α that previously worked well. Nevertheless, we were able to observe a faint band for the lower isoform of DNMT3A in AGO1 IP from nuclear extract of WT mESCs (Revised Fig EV1C), however it is important to note that as most of the AGO1 is cytoplasmic we probably IP a very small fraction of AGO1 in such experiment.

Due to our failure to validate MS data with Co-IP followed by WB, we now removed this MS data from our revised manuscript.

3. Reviewer 1 asked us whether changes observed in figures 2-4 are due to a direct effect of AGO1 or the global effects of disrupting AGO1 expression on gene expression.

The analysis of the RNA-seq data for *Ago1*_KO compared to WT mESCs is available in (Mueller *et al*, 2021). This analysis demonstrates that global RNA levels are only mildly impacted upon *Ago1* deletion, thus the observed redistribution of H3K9me3, does not seem to have a global impact on gene expression. We now cite this study in our revised manuscript.

4. Reviewer 2 asked us to study the major satellite transcript levels under rescue conditions.

In our H3K9me3 IF in the AGO1 complemented polyclonal population, we performed an HA co-staining to be able to select the cells that actually express AGO1 for the analysis. However, this co-staining did not work well for us with the RNA FISH protocol, due to the nuclear pre-extraction procedure necessary to visualize major satellites properly. This led to the problem that we could not distinguish within the population which cells actually re-express AGO1 at appropriate levels. Therefore, we newly generated two AGO1 complemented single clones (*Ago1*_KO + 3x HA-AGO1 A9, B2), with a good AGO1 expression (Revised Fig S2). We have integrated the qPCR and RNA FISH data on major satellites in the AGO1 rescued single clones (Revised Fig 3 and S3). These data show that the reintroduction of AGO1 can rescue partially major satellite levels, reinforcing our original findings.

5. All reviewers asked us about the specificity of the miRNAs identified to target major satellite transcripts. We have shown that using miRNA inhibitors against miR-30 a, d, e-3p increased major satellite levels.

We have now added an additional experiment to our revision, where we performed the opposite approach (Revised Fig 4F). In this experiment we transfected WT mESCs with miRNA mimics (miR-30e-3p) and observed that adding more miR-30e-3p to the cells decreases the major satellite transcript levels, which is the opposite of what we have observed with the inhibitors. These two independent lines of data reinforce the idea that miR-30e-3p is involved in major satellites regulation.

In addition, to reinforce the hypothesis that miRNAs are involved in the regulation of pericentromeric regions in mESCs, we now present, in our revised manuscript, IFs of H3K9me3 and HP1 α showing a redistribution of these proteins away from the pericentromeres also in *Droscha_KO* mESCs (see above and Fig 4A and B).

6. Reviewer 3 ask us to show the effect of miR-30 inhibitors on H3K9me3 deposition at pericentromeric foci.

We have already performed this experiment and included it in this rebuttal letter. These results show a mild and statistically not significant decrease of H3K9me3 upon transient miR-30-3p inhibitor transfection, but again here one must keep in mind that mESCs are difficult to transfect (around 50% efficiency) and since we cannot reliably distinguish cells within the population cells that are truly transfected the observed phenotype might be an underestimation.

7. Reviewer 3 mentioned that our working model is rather speculative.

We agree with this statement and we have now deleted it from our revised manuscript and added more explanations in our discussion.

1. If the editor agrees we will not monitor the distribution of H3K9me3 and HP1 α in *Ago2_KO* mESCs, as we now want to refocus the entire paper on AGO1 as requested by the reviewers.

2. We prefer not to perform the rescue experiments with and AGO1 RNA binding mutant. The structure of AGO1 and its RNA binding domain is only available for human AGO1, so it would need to be verified and validated in mouse AGO1. Furthermore, RNA binding mutant AGOs are often unstable (at least this was what we observed for AGO2, (Ngondo *et al*, 2018)) making it more challenging to perform such experiments.

Other data that we want to add to the rebuttal letter, but decided to NOT include in our revised manuscript:

Instead of using an AGO1-RNA binding mutant, we complemented our *Ago1*_KO mESCs with a plasmid containing a tagged-AGO1 with a nuclear export signal (NES) to assess if only the cytoplasmic function of AGO1 will be required to restore the localization of H3K9me3 and HP1 α to the pericentromeric regions (see below Figure 4A).

Figure 4: NES-HA-AGO1 rescue experiments at pericentromeric regions. A) Schematic representation of the experimental design to reintroduce NES-HA-AGO1 in *Ago1*_KO mESCs. B) Representative IF images of H3K9me3 in WT, *Ago1*_KO and a representative image of *Ago1*_KO + NES-HA-AGO1 mixed population mESCs. scale bar = 10 μ m. Right: Quantification of foci count for H3K9me3 that colocalizes with DAPI regions in WT, *Ago1*_KO and *Ago1*_KO + NES-HA-AGO1 mixed population mESCs. The graph shows the median distribution with the interquartile range. **** = pvalue < 0.0001, * = pvalue < 0.05, Mann-Whitney test for n=3 independent experiments. C) Western

blots for HA and AGO1 in WT, *Ago1_KO* and *Ago1_KO* + NES-HA-AGO1 single clones. D) Western blots for the Cytoplasmic/Nuclear fractionation to visualize AGO1 in WT and the two single clones *Ago1_KO* + NES-HA-AGO1 B1 and B4. TUBULIN (cytoplasm), H3K9me3 (chromatin) were used as subcellular markers.

With this approach we were able to see a partial rescue of the phenotype in polyclonal population of cells (Fig 4B). In order to verify the strict localization of the NES-AGO1 in the cytoplasm of the complemented mESCs, we then generated single clones (Fig 4C) and performed nuclear/cytoplasmic fractionation followed by WB. Here, we observed that despite the NES tag, AGO1 was present in both nuclear and cytoplasmic fractions, suggesting again an active mechanism to transport AGO1 in the nucleus and not allowing us to conclude about a nuclear role of AGO1 in the rescue of the phenotype. In light of these results, we decided to not include these new experiments in our revised version of the manuscript.

Additionally, in order to assess the presence of AGO1 at pericentromeric regions, we previously try to visualize nuclear AGO1. However, due to the low amount of AGO1 within the nucleus it was not possible to see such a defined co-localization by a nuclear pre-extraction followed by an IF (Fig 5). We have prepared some IF images of HA-AGO1 localization in the cell line: *Ago1_KO* + 3x HA-AGO1 mixed population and are happy to share them here:

Fig. 5 HA-AGO1 expression in complemented cell lines. On the left side: an IF for HA is depicted in WT and *Ago1_KO* + 3x HA-AGO1 mixed population. WT cells should not express HA and therefore serve as a control. A green fluorescent secondary antibody was used. The nuclear signal seems slightly stronger for the complemented cell line, however in general it is rather weak. On the right side: an IF for HA is depicted in WT and *Ago1_KO* + 3x HA-AGO1 mixed population. WT cells should not express HA and therefore serve as a control. A red fluorescent secondary antibody was used. The nuclear signal again seems very weak.

We also performed an HA-AGO1 ChIP followed by qRT-PCR.

Figure 6. HA pull-down at major satellites. ChIP-qPCR results for two independent major satellite primer pair sets in WT and the two single clones Ago1_KO + HA-AGO1 A9 and B2.

In WT there is no HA tag, so we should see no enrichment, therefore it also serves as a negative control (Fig 6). However, the HA pull-down in the A9 and B2 complemented clones compared to WT was not much stronger or absent. It is hard to detect HA AGO1 at major satellites via ChIP-qPCR.

Using these techniques we were unable to locate AGO1 at the pericentromeric regions in mESCs, despite confirming nuclear localization of AGO1 in mESCs by fractionation experiments followed by western blotting (Fig 4C of our revised manuscript).

3. We also prefer not to perform chromatin bound AGO1-IP followed by RNA-seq. Although this is a very good point, it is a rather tricky experiment and would take longer than three months. We have observed very few AGO1 on chromatin (as shown in Fig 1A), so this would be hard to prove. Also, to perform AGO1-nuclear IP followed by small RNA-seq and analysis would require us more time than three months.

February 14, 2022

RE: Life Science Alliance Manuscript #LSA-2021-01277-TR

Prof. Constance Ciaudo
ETH Zurich
D-Biol
IMHS HPL G32.1
Otto-Stern-Weg 7
Zurich 8093
Switzerland

Dear Dr. Ciaudo,

Thank you for submitting your revised manuscript entitled "AGO1 regulates pericentromeric regions in mouse embryonic stem cells". We would be happy to publish your paper in Life Science Alliance pending final revisions necessary to meet our formatting guidelines.

- please add your main, supplementary figure, and table legends to the main manuscript text after the references section
- please make sure the manuscript sections are aligned per LSA's formatting guidelines: please separate the Figure legends and Supplemental Figure legends into separate sections
- in the Data Availability Statement, please remove Reviewer access tokens, but leave the accession numbers and make them publicly accessible

A. FINAL FILES:

B. MANUSCRIPT ORGANIZATION AND FORMATTING:

Sincerely,

February 17, 2022

RE: Life Science Alliance Manuscript #LSA-2021-01277-TRR

Prof. Constance Ciaudo
ETH Zurich
D-Biol
IMHS HPL G32.1
Otto-Stern-Weg 7
Zurich 8093
Switzerland

Dear Dr. Ciaudo,

Thank you for submitting your Research Article entitled "AGO1 regulates pericentromeric regions in mouse embryonic stem cells". It is a pleasure to let you know that your manuscript is now accepted for publication in Life Science Alliance. Congratulations on this interesting work.

DISTRIBUTION OF MATERIALS:

Again, congratulations on a very nice paper. I hope you found the review process to be constructive and are pleased with how the manuscript was handled editorially. We look forward to future exciting submissions from your lab.

Sincerely,
